# IFNγ suppresses the expression of GFI1 and thereby inhibits Th2 cell proliferation

**Murshed H. Sarkar**[1�], **Ryoji Yagi**[1�]*, **Yukihiro Endo**[1,2], **Ryo Koyama-Nasu**[1,2], **Yangsong Wang**[1,2], **Ichita Hasegawa**[1,2], **Toshihiro Ito**[1], **Ilkka S. Junttila**[3,4], **Jinfang Zhu**[5], **Motoko Y. Kimura**[1,2]*, **Toshinori Nakayama**[1]

**1** Department of Immunology, Chiba University, Inohana, Chuo-ku, Chiba, Japan, **2** Department of Experimental Immunology, Chiba University, Inohana, Chuo-ku, Chiba, Japan, **3** Faculty of Medicine and Health Technology, Tampere University, Tampere, Finland, **4** Fimlab Laboratories, Pirkanmaa Hospital District, Tampere, Finland, **5** Laboratory of Immunology, National Institute of Allergy and Infectious Diseases, National Institutes of Health, Bethesda, Maryland, United States of America

যω These authors contributed equally to this work.
* ryojiyagi@gmail.com (RY); kimuramo@chiba-u.jp (MYK)

**Data Availability Statement:** All relevant data are within the manuscript and its Supporting information files.

## Abstract

While IFNγ is a well-known cytokine that actively promotes the type I immune response, it is also known to suppress the type II response by inhibiting the differentiation and proliferation of Th2 cells. However, the mechanism by which IFNγ suppresses Th2 cell proliferation is still not fully understood. We found that IFNγ decreases the expression of growth factor independent-1 transcriptional repressor (GFI1) in Th2 cells, resulting in the inhibition of Th2 cell proliferation. The deletion of the *Gfi1* gene in Th2 cells results in the failure of their proliferation, accompanied by an impaired cell cycle progression. In contrast, the enforced expression of GFI1 restores the defective Th2 cell proliferation, even in the presence of IFNγ. These results demonstrate that GFI1 is a key molecule in the IFNγ-mediated inhibition of Th2 cell proliferation.

## Introduction

Naïve CD4 T cells that recognize foreign antigens through their T cell receptors (TCRs) differentiate into several effector T helper (Th) subsets, such as Th1, Th2 and Th17 cells, according to the surrounding cytokine environments, which are induced by specific pathogens [1–5]. Each Th subset produces distinct signature cytokines (i.e. IFNγ and TNFβ from Th1 cells; IL-4, IL-5 and IL-13 from Th2 cells; IL-17A and IL-17F from Th17 cells), which leads to unique immune responses, by activating and recruiting a variety of immune cells at the site of inflammation. During infection, a select subset of effector Th cells efficiently expands and creates an appropriate immune response by producing cytokines that promote the differentiation of newly antigen-recognized CD4 T cells toward the same specific type of Th cells but also inhibiting the differentiation and proliferation of other types of Th cells [6–8]. Consequently, the newly activated CD4 T cells converge to differentiate into specific types of effector subsets to create an environment that supports appropriate immune responses.

**Funding:** This work was supported by Leading Graduate School at Chiba University, Nurture of Creative Research Leaders in Immune System Regulation and Innovative Therapeutics), and by the following grants: Ministry of Education, Culture, Sports, Science and Technology (MEXT Japan) Grants-in-Aid for Scientific Research (S) 26221305, JP19H05650, (B) JP20H03464, (C) 15K08523, 15K08524, 18K07165, Research activity Start-up 25893033, the Kanae Foundation for the Promotion of Medical Science, the Kashiwado Memorial Foundation, the Hamaguchi Foundation for the Advancement of Biochemistry, the Takeda Science Foundation, Academy of Finland 25013080481, National Academy of Finland 2501342041, Competitive State Research Financing of the Expert Responsibility Area of Fimlab Laboratories X51409, Tampere University Hospital Support Foundation and Tampere Tuberculosis Foundation. The funders had no role in study design, data collection and analysis, decision to publish, or preparation of the manuscript.

**Competing interests:** The authors declare no conflicts of interest in association with the present study.

IFNγ, a signature cytokine produced by Th1 cells, induces the expression of T-bet, encoded by the *Tbx21* gene, in activated CD4 T cells [9]. T-bet binds to the regulatory elements on the *Ifng* gene locus, and induces chromatin remodeling through histone modification so that some transcription factors can access the gene locus for *Ifng* transcription [10, 11]. Thus, during the type I immune response, Th1 cells expand via a positive feedback loop through the IFNγ-T-bet axis. In contrast, IFNγ inhibits the proliferation and differentiation of other Th cells, such as Th2 and Th17 cells [7, 8, 12, 13]. For instance, it has been reported that IFNγ directly inhibits Th2 cell proliferation [14–16] and that the IFNγ-induced T-bet expression suppresses Th2 cell differentiation by inhibiting the expression and function of GATA3 [17, 18]. Furthermore, IFNγ suppresses Th17 cell proliferation by inhibiting the expression of IL-23R [19], and the IFNγ-induced T-bet expression also suppresses Th17 cell differentiation through the inhibition of the Runx1-mediated transactivation of Rorc (which encodes the transcription factor RORγt) by making a complex of T-bet and Runx1 [20]. This evidence strongly suggests that IFNγ not only enhances the type I immune response but also inhibits other immune responses (i.e., Th2 and Th17 cells), resulting in the efficient differentiation of CD4 T cells into Th1 cells. However, the molecular mechanism by which IFNγ inhibits Th2 cell-mediated immune responses remains poorly understood.

Murine GFI1 is a transcriptional repressor whose protein size is about 45 kDa expressed in hematopoietic-derived cells. The N-terminal region of GFI1 contains a SNAG domain that serves to recruit proteins that modify histones and is present in other transcriptional repressors. The C-terminal region of GFI1 contains six $C_2H_2$-type zinc fingers that are essential for DNA binding. GFI1 recruits chromatin-modifying enzymes to modify the chromatin accessibility for gene transcription in various biological settings. GFI1 plays a crucial role in the development and function of hematopoietic stem cells, B and T cells, dendritic cells, granulocytes and macrophages [21], suggesting its crucial role in various types of cells.

In this study, we explored the mechanism underlying the IFNγ-mediated inhibition of Th2 cell proliferation, in which IFNγ decreased the expression of *Gfi1*, resulting in a delay in Th2 cell proliferation. The loss of GFI1 reduced Th2 cell proliferation, whereas the enforced expression of GFI1 in IFNγ-treated Th2 cells restored the defective proliferation. These data suggest that the IFNγ-mediated inhibition of the *Gfi1* gene expression leads to efficient Type I responses by suppressing Th2 cell proliferation.

## Materials and methods

### Mice

*Gfi1*^fl/fl mice [22] were bred with CreER^T2 (Taconic) transgenic mice to generate the *Gfi1*^fl/fl CreER^T2 line. C57BL/6 mice and CD45.1 congenic mice were purchased from Clea Japan and Sankyo Laboratory, respectively. *Tbx21*^-/- mice were kindly provided by Dr. Laurie Glimcher (Cornell University) [23]. All mice used in this study were maintained under specific-pathogen-free conditions. All experimental protocols using mice were approved by the Chiba University Animal Committee. All animal care was performed in accordance with the guidelines of Chiba University.

### Preparation of cells

For naïve CD4 T cell purification, CD4 T cells isolated from lymph nodes and/or spleen were first enriched with anti-CD4 microbeads (Miltenyi Biotec) followed by positive selection using autoMACS (Miltenyi Biotec), and then naïve CD44^lo CD62L^hi CD4 T cells were sorted by FACSAria (BD). The purity was more than 98%. T cell-depleted splenocytes were prepared by incubation with anti-Thy1.2 FITC monoclonal antibody (mAb) (BioLegend) and anti-FITC

microbeads (Miltenyi Biotec), followed by negative selection using autoMACS. T cell-depleted splenocytes were irradiated at 30.67 Gy and used as antigen-presenting cells, as previously described [24].

### *In vitro-differentiated Th* culture and IFNγ treatment

Naïve CD4 T cells ($2x10^5$ cells) were stimulated with 1 µg/ml of anti-CD3 and 1 µg/ml of anti-CD28 Abs together with irradiated T cell-depleted splenocytes in the presence of various combinations of antibodies (Abs) and cytokines for 2 or 3 days: for Th1-polarizing conditions, 10 ng/ml IL-12, 5 ng/ml IL-2 and 10 µg/ml anti-IL-4; and for Th2-polarizing conditions, 10 ng/ml IL-4, 5 ng/ml IL-2 and 10 µg/ml anti-IFNγ. These cells were further cultured in the medium containing 5 ng/ml IL-2- for another 2 or 3 days. *In vitro*-differentiated Th1 and Th2 cells were used for further experiments.

For the IFNγ treatment, *in vitro*-differentiated Th2 cells ($2.5x10^5$ cells/ml) were cultured with IL-2 (5ng/ml) in the presence or absence of IFNγ (10 ng/ml or indicated concentrations) for the days indicated in rhe figure legends. For the inhibition of STAT1 activation by using fludarabine, *in vitro* differentiated Th2 cells were further cultured in IL-2-containing medium with or without IFNγ in the presence or absence of fludarabine (100µM) for 3 days.

### Cell proliferation analyses

*In vitro*-differentiated Th cells were washed with PBS and then labelled with CFSE (10 µM) according to the manufacturer's protocol (Thermo Fisher Scientific). Cell proliferation was determined by both cell counting and the mean fluorescence intensity of CFSE dilution on cells by flowcytometory analysis.

### Cell cycle analyses

The cell cycle status was measured by BrdU staining according to the manufacturer's protocol (BD). Cells were treated with BrdU (10 µM) for 4 hours before harvesting. The cell cycle status was determined by BrdU and 7-AAD staining.

### Detection of cell death

*In vitro* cultured cells were first stained with antibodies for the cell surface molecules, washed twice with cold PBS and then treated with FITC-conjugated Annexin V and PI for 15 min at room temperature (RT) according to the manufacturer's protocol (BD). Cell death was determined by Annexin V and PI staining based on flowcytometory.

### RNA purification and quantitative RT-PCR

Total RNA was isolated using TRIzol (Thermo Fisher Scientific), and cDNAs were prepared using SuperScript II Reverse Transcriptase (Thermo Fisher Scientific). Quantitative PCR was performed on a 7500 real-time PCR system (Applied Biosystems) using the predesigned primer-probe sets listed in S1 Table.

### Retroviral construction and infection

*Gfi1* cDNA was cloned from the total RNA of *in vitro*-differentiated Th2 cells. After sequencing, the cloned *Gfi1* cDNA was inserted into Thy1.1-Retrovirus (RV) vector. Retroviruses were prepared from the culture supernatants of the Plat-E packaging cells transfected with RV constructs using TransIT-LT1 Transfection Reagent (Mirus), and the culture supernatants were centrifuged at 12,000 ×*g* for 14–18 hours at 4˚C in order to concentrate RV. *In vitro-*

differentiated Th2 cells ($2.5 \times 10^5$ cells) were cultured in the presence of IL-2 with concentrated RV mixed with Polybrene Transfection Reagent (final concentration: 8 ug/mL; Millipore) and subjected to spin-infection by centrifugation at 2,500 rpm for 1 hour at 24°C [25].

## 4-OHT treatment

*In vitro*-differentiated Th2 cells were cultured in the medium containing 5 ng/ml IL-2 with 500 nM 4-OHT (Sigma) for the days indicated in the figure legends.

## Intracellular staining

As previously described [26], the activated CD4 T cells were restimulated with 10 ng/ml phorbol 12-myristate 13-acetate (PMA) and 500 nM ionomycin in the presence of 2 μM monensin for 4 hours, then stained with anti-CD44 and anti-CD4 Abs on ice for 30 min, fixed with 4% paraformaldehyde at room temperature for 10 min and permeabilized for 10 min on ice with PBS containing 0.5% Triton X-100 and 0.1% BSA. After permeabilization, the cells were stained with anti-cytokine Abs and assessed by FACS Canto II (BD). The FlowJo software program (Tree Star) was used to analyze the data. Abs specific to mouse CD4, CD44, IL-4, IL-5, IL-13, and IFNγ were purchased from BioLegend.

## Statistical analyses

Unless otherwise indicated, p values were calculated using Student's *t*-test.

## Results

### IFNγ inhibits Th2 cell proliferation due to the blockade of G1/S progression

To examine the molecular mechanism through which IFNγ inhibits Th2 cell proliferation, we set up an *in vitro* culture system, in which *in vitro* fully differentiated Th2 cells (S1A Fig) were labelled with CFSE, and further cultured with IL-2 in the presence or absence of IFNγ for 5 days (S1B Fig). After day 3, the CFSE expression in Th2 cells treated with IFNγ was significantly higher than in untreated Th2 cells (Fig 1A and 1B), and the number of Th2 cells treated with IFNγ was significantly decreased (Fig 1C), suggesting that IFNγ treatment inhibits Th2 cell proliferation in this *in vitro* culture system.

To examine the sensitivity of the IFNγ concentration that inhibits Th2 cell proliferation, we next used different concentrations of IFNγ (S1C Fig). The suppressive impact of IFNγ treatment on Th2 cell proliferation was observed even with 0.3 ng/ml. Since the concentration of 10 ng/ml showed a plateau with regard to inhibition, we decided to use 10 ng/ml for subsequent experiments. We also confirmed the same results using OTII TCR transgenic (Tg) Th2 cells generated with specific antigen stimulation (i.e. chicken ovalbumin peptide 323–329) instead of polyclonal stimulation (i.e. anti-CD3 and anti-CD28 Abs) (S1D–S1G Fig), indicating that the IFNγ-mediated inhibition of Th2 cell proliferation occurred in both polyclonal and monoclonal Th2 cells.

We further found that IFNγ treatment significantly decreased the frequency of cells in the S phase. Although the frequency of cells in the G2-M phase was slightly increased (Fig 1D and 1E, and S1H and S1I Fig), this was likely a secondary effect due to the reduced number of cells. We also examined the impact of IFNγ treatment on death among Th2 cells (Fig 1F). Cell death was slightly increased in Th2 cells with IFNγ treatment; however, the frequency of dead cells was less than 3% at all time points, suggesting that cell death is not major effect of IFNγ

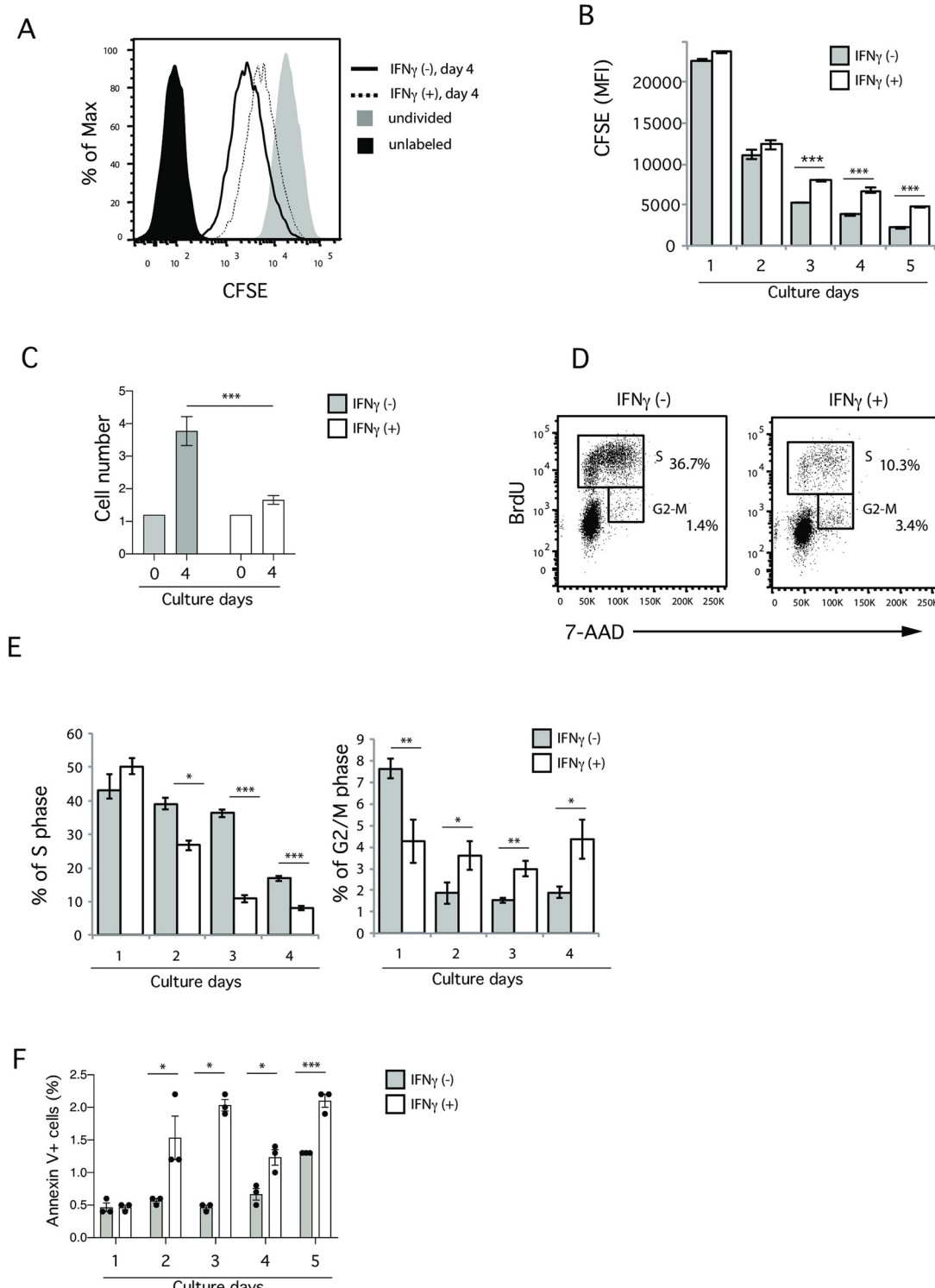

**Fig 1. IFNγ inhibits Th2 cell proliferation.** (A) Flow cytometry showing the mean fluorescence intensity (MFI) of CFSE dilution on day 4 of IFNγ treatment. (B) The MFI of Th2 cells cultured in the presence or absence of IFNγ on the indicated number of days. (C) The cell number was determined after 4 days of culture of IL-2 in either the presence or absence of IFNγ. (D) A cell cycle analysis of Th2 cells after 4 days of culturing in the presence or absence of IFNγ. BrdU incorporation (final four hours) was measured to determine the distribution of cells at each stage of the cell cycle. (E) The frequencies of cells in the S phase (left) and G2/M phase (right) on the indicated number of days are shown. (F) Annexin V+ cells were determined on the indicated days of IFNγ treatment. Data are representative of three independent experiments. Error bars represent the mean ± SD. *, **, and *** $p < 0.05$, $p < 0.01$, and $p < 0.001$, respectively (Student's *t*-test).

treatment causing defects in Th2 cell proliferation. These results suggest that the IFNγ-mediated inhibition of cell proliferation likely occurred due to the blockade of G1/S progression.

Since Th cell proliferation is highly dependent on IL-2 signalling, one possible mechanism by which IFNγ inhibits Th2 cell proliferation involves the suppression of IL-2 receptor expression by IFNγ treatment. However, this possibility is unlikely, as the expression levels of γC (CD132) was comparable between IFNγ-treated and untreated cells, and the CD25 and CD122 expression was increased by IFNγ treatment (new S1J Fig).

## T-bet is not involved in IFNγ-mediated inhibition of Th2 cell proliferation

To identify the molecules responsible for the IFNγ-mediated inhibition of Th2 cell proliferation, we first examined the possibility that the T-bet expression induced by IFNγ inhibited Th2 cell proliferation, since T-bet is known to suppress Th2 cell differentiation (17, 18) and might influence Th2 cell proliferation. To test this, *in vitro*-differentiated Th2 cells from *Tbx21*⁻/⁻ (CD45.2) and *Tbx21*⁺/⁺ (CD45.1) mice were mixed together, labelled with CFSE, and cultured in the presence of IL-2 with or without IFNγ for the indicated number of days (S2A Fig). We first confirmed that the expression of T-bet mRNA was induced upon IFNγ stimulation (S2B Fig), and that Th cells from *Tbx21*⁻/⁻ mice showed a significant reduction in their IFNγ production due to the lack of T-bet expression (new S2C Fig). However, contrary to our expectation, IFNγ treatment efficiently inhibited the proliferation of *Tbx21*-deficient Th2 cells (new S2D Fig). Furthermore, Th1 cells expressing high T-bet levels did not show any proliferative defects due to IFNγ treatment (new S2E Fig). These results indicate that the IFNγ-induced T-bet expression is not responsible for the inhibition of Th2 cell proliferation.

## GFI1 deficiency dampens Th2 cell proliferation

To identify the molecules involved in the IFNγ mediated inhibition of Th2 cell proliferation, we examined the gene expressions in Th2 cells with and without IFNγ treatment. We focused on 37 transcription factors reported to be involved in the growth, development, effector function, apoptosis, and survival of helper T cells as listed in S3A Fig. IFNγ-treated Th2 cells had higher expressions of *Tbx21*, *Irf1*, *Helios* and *Atf3* genes than IFNγ-untreated Th2 cells, whereas the *Foxp3*, *Gfi1* and *Sox4* genes were downregulated in IFNγ-treated Th2 cells (S3B Fig). Because *Helios* and *Sox4* have been reported to not influence Th2 cell proliferation [27, 28], and *Tbx21* was not involved in IFNγ-mediated inhibition of Th2 cell proliferation (S2 Fig), we decided to focus on the GFI1 molecule, which has been shown to be highly expressed in Th2 cells [29] and play important roles in Th2 cell differentiation [30]. Importantly, we found that the GFI1 mRNA was significantly decreased in IFNγ-treated Th2 cells (Fig 2A). In addition, we found that mRNA expression of *Cdkn1*, a cell cycle inhibitor known to be a target of GFI1 [31], was significantly increased by IFNγ treatment (new Fig 2B). Accordingly, we hypothesized that the IFNγ-mediated downregulation of GFI1 might result in reduced Th2 cell proliferation.

To test this idea, we udrf *Gfi1*ᶠˡ/ᶠˡ CreERᵀ² mice, which allowed us to remove the *Gfi1* gene from fully differentiated Th2 cells by treating them with 4-hydroxytamoxifen (4-OHT) (new S4A Fig), thereby avoiding any influence from defects of Th2 cell differentiation due to GFI1 deficiency. We confirmed the substantial deletion of GFI expression by 4-OHT treatment using RT-PCR (new S4B Fig). Furthermore, we confirmed that the *in vitro* Th2 cell differentiation was comparable between *Gfi1*ᶠˡ/ᶠˡ CreERᵀ² and *Gfi1*⁺/⁺ CreERᵀ² mice, as they showed a similar ability to produce IL-4, IL-5 and IL-13 (new S4C Fig). *In vitro* fully differentiated Th2 cells from either *Gfi1*ᶠˡ/ᶠˡ CreERᵀ² (CD45.2+) or *Gfi1*⁺/⁺ CreERᵀ² (CD45.2+) mice were mixed together with wild-type Th2 cells (CD45.1+) as an internal control, labelled with CFSE and

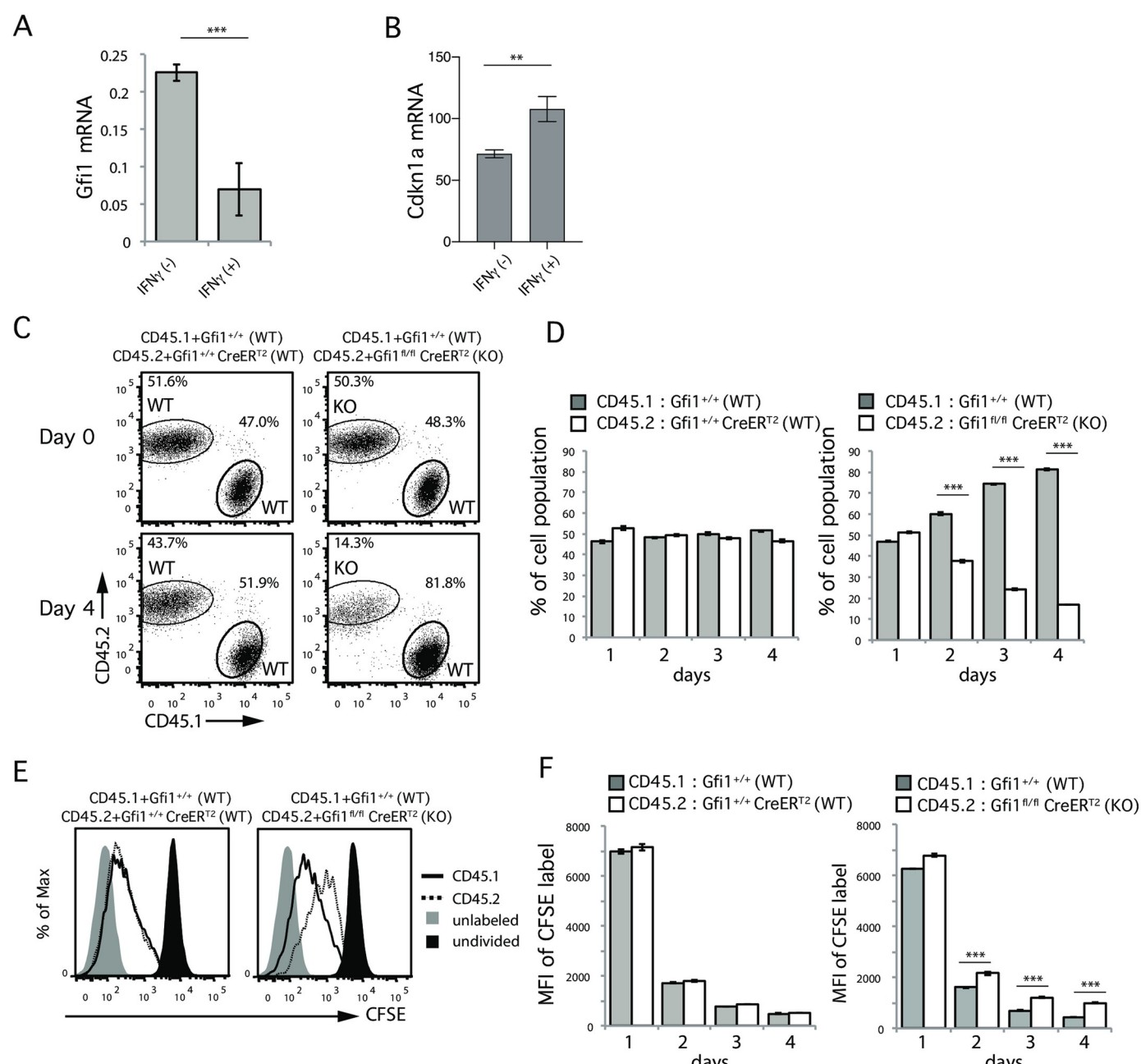

**Fig 2. Loss of *Gfi1* leads to impaired Th2 cell proliferation.** (A) The *Gfi1* mRNA expression relative to *Hprt* on Th2 cells after treatment with or without IFNγ for 2 days. Error bars represent the mean ± SD. Data are representative of three independent experiments. (B) The *Cdkn1a* mRNA expression relative to *β-actin* on Th2 cells after treatment with or without IFNγ for 4 days. (C and D) Relative cell expansion of *Gfi1*^fl/fl^ CreER^T2^ (KO, CD45.2) or *Gfi1*^+/+^ CreER^T2^ (WT, CD45.2) Th2 cells in comparison to CD45.1 WT Th2 cells on day 4 of culturing in the presence of 4-OHT (C). The percentages of CD45.2 (WT or KO) or CD45.1 (WT) Th2 cells were measured on the indicated days (D). (E and F) Histogram of the CFSE expression on Th2 cells on day 4 of culturing in the presence of IFNγ (E). The MFI of CFSE on Th2 cells on the indicated days of culture in the presence of IFNγ was determined (F). *** $p < 0.001$ (Student's *t*-test).

then cultured with IL-2 in the presence of 4-OHT for the indicated number of days (new S4A Fig). We found that the frequency of *Gfi1*-deficient Th2 cells (CD45.2) was significantly decreased after four days of culture with IL-2 (new Fig 2C right), whereas the frequency of *Gfi1*-sufficient Th2 cells (CD45.2) was comparable to that of CD45.1 internal control Th2 cells

(new Fig 2C left), demonstrating that GFI1 deficiency resulted in less marked proliferation of Th2 cells in a cell-intrinsic manner.

A time course analysis showed that the impact of GFI1 deficiency on Th2 cell proliferation was significant at later time points (new Fig 2D). Consistent with this result, the CFSE expression by *Gfi1*-deficient Th2 cells was significantly higher than by *Gfi1*-sufficient Th2 cells (new Fig 2E and 2F). These results indicate that GFI1 deficiency results in less marked proliferation of Th2 cells.

## GFI1 deficiency results in the blockade of G1/S progression

We next examined whether *Gfi1* deficiency had any impact on the progression of the cell cycle. We found that the frequency of *Gfi1*-deficient Th2 cells in the S phase was significantly decreased, while the frequency of *Gfi1*-sufficient Th2 cells in the S phase was similar to that of CD45.1+ control Th2 cells (Fig 3A and 3B). Although the frequency of *Gfi1*-deficient Th2 cells in the G2/M phase was significantly increased, we think that this was a secondary effect due to the reduced number of Gfi1-deficient Th2 cells. In fact, we detected less cell recovery of Gfi1--deficient Th2 cells than of Gfi1-sufficient Th2 cells in the same culture (Fig 3A bottom left, 27.3% vs. 71.9%), and we noted no marked differences in the frequency of cells in the G2/M phase when we calculated the frequency of cells in each phase within the same culture (Fig 3C). These results demonstrate that GFI1 deficiency results in the blockade of cell cycle progression, inhibiting cell proliferation. Notably, these phenotypes observed in GFI1-deficient Th2 cells were very similar to those in IFNγ-treated Th2 cells (Fig 1), suggesting that the IFNγ-dependent inhibition of Th2 cell proliferation is mediated by the downregulation of the GFI1 expression.

Since IFNγ signaling activates STAT1, we next examined if the inhibition of STAT1 activation using a STAT1 inhibitor (Fludarabine) rescued the IFNγ-mediated inhibition of Th2 cell proliferation. Interestingly, we noted no rescue of proliferation, since the MFI of CFSE on IFNγ-treated Th2 cells did not change between the presence or absence of Fludarabine (new Fig 3D left), whereas cell number of IFNγ-treated cells in the presence of Fludarabine (new Fig 3D right) was seemed to be slightly rescued although it was not significant. These data suggest that the IFNγ-mediated inhibition of Th2 cell proliferation was not regulated by STAT1 activation.

## The ectopic expression of GFI1 restores the IFNγ-dependent inhibition of Th2 cell proliferation

To examine whether the IFNγ-dependent inhibition of Th2 cell proliferation was due to the loss of GFI1 expression, we retrovirally introduced GFI1 into IFNγ-treated Th2 cells and examined the effect of the introduction of GFI1 on the restoration of defective Th2 cell proliferation, including in the presence of IFNγ (new S4D Fig). The efficiency of retrovirus infection with the GFI1-retrovirus and control Thy1.1-retrovirus was almost identical, regardless of IFNγ treatment (new S4E Fig). Notably, the introduction of GFI1 substantially restored the defective Th2 cell proliferation, even in the presence of IFNγ (new Fig 3E). These results demonstrate that IFNγ-mediated GFI1 downregulation is a key to dampening Th2 cell proliferation.

## Discussion

The present study reveals the molecular mechanism through which IFNγ inhibits Th2 cell proliferation. We found that IFNγ decreases the expression of transcription factor GFI1, and *Gfi1*-deficient Th2 cells show defective proliferation due to the inhibition of G1/S progression.

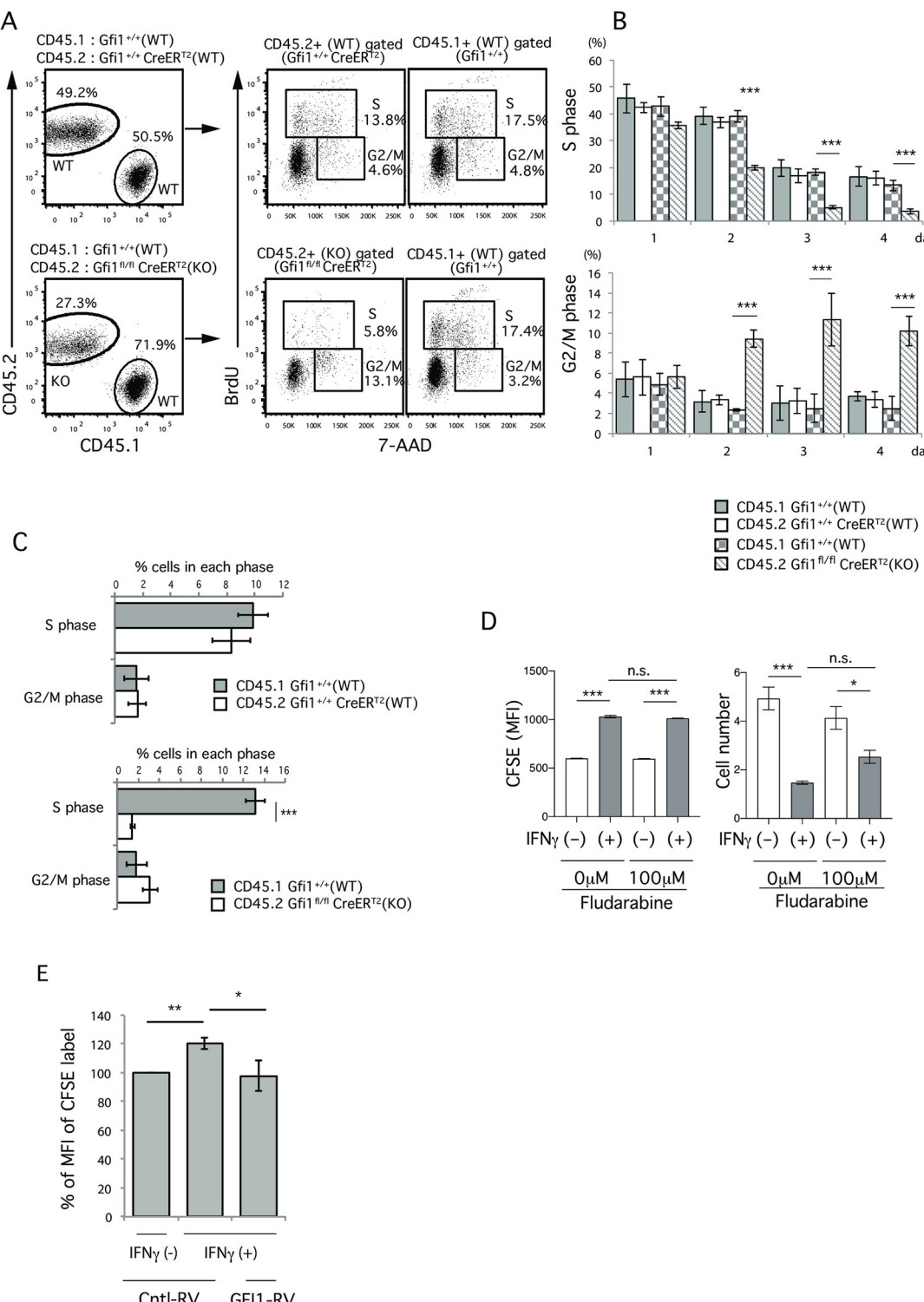

**Fig 3. The loss of *Gfi1* results in the blockade of G1/S, and the ectopic expression of GFI1 restores the IFNγ-mediated inhibition of Th2 cell proliferation.** (A-C) The relative cell expansion of *Gfi1*^fl/fl^ CreER^T2^ (KO, CD45.2) or *Gfi1*^+/+^ CreER^T2^ (WT, CD45.2) Th2 cells in comparison to CD45.1 WT Th2 cells on day 4 of culturing in the presence of IFNγ was determined (A, left). A cell cycle analysis was performed after gating of the indicated cell populations (A, right). The percentages of cells in the S phase or G2/M phase after gating of the indicated cell populations are shown (B). The percentages of cells in S phase or G2/M phase of the indicated cell populations without the gating of CD45.1 or CD45.2+ cells (i.e., % in the culture) on day 3 (C). (D)

The MFI of CFSE on Th2 cells (left) and cell number (right) after the cultivation with or without IFNγ in the presence or absence of Fludarabine (100μM) for 3 days was shown. (E) The MFI of CFSE on Th2 cells that were introduced to either GFI1-Thy1.1 RV or Control-Thy1.1 RV after 4 days of culturing in the presence or absence of IFNγ (protocol in new S4D Fig). The % of MFI of CFSE on Th2 cells relative to the MFI of CFSE on Th2 cells introduced control-Thy1.1-RV in the absence of IFNγ treatment. Data are representative of three independent experiments. Error bars represent the mean ± SD. *, **, and *** *p* < 0.05, *p* < 0.01, and p < 0.001, respectively (Student's *t*-test).

Furthermore, enforced GFI1 expression restored the IFNγ-mediated inhibition of Th2 cell proliferation. These results demonstrate that IFNγ-mediated inhibition of the GFI1 expression is key to the IFNγ mediated inhibition of Th2 cell proliferation.

IFNγ is a well-known Type I cytokine important for the promotion of Type I immune responses. Our data suggest that IFNγ treatment inhibits the expression of GFI1, which is important for suppressing Th2 cell expansion and promoting efficient Type-I immune responses. While GFI1 is reported to be highly expressed in Th2 cells [29] and plays important roles in Th2 cell differentiation [30], GFI1 works not only in Th2 cells but also in other types of helper T cells, including Th1 and Th17 cells, despite the low GFI1 expression in these cells. In fact, it has been reported that GFI1-deficient CD4T cells showed enhanced responses driven by other types of cells, such as Th1 and Th17 cells [29, 32, 33]. *Gfi1*-deficient Th1 cells intrinsically enhance Th1 cytokines [33], and *Gfi1*-deficient Th17 cells increase the percentages of IL-17A-producing cells to a greater extent than wild-type Th17 cells [29, 32]. IFNγ-dependent GFI1 downregulation can occur in Th1 cells, which is expected to lead to efficient Type-I immune responses. Indeed, we previously reported that GFI1 represses IFNγ gene activation, suggesting that IFNγ represses the expression of GFI1, which further increases the production of IFNγ to lead to efficient Type-I immune responses [34].

Our data showed that both IFNγ treatment and GFI1 deficiency dramatically decreased the frequency of G1-S cells and increased the frequency of G2-M cells (Figs 1D, 1E, 3A and 3B), suggesting that GFI1 deficiency either inhibits G1/S progression or specifically increases G2-M cells. However, our cell count data (Fig 1C) and co-culture system seen in Fig 3C clearly showed that the inhibition of G1-S progression was a primary effect, while the accumulation of G2-M was a secondary effect induced by GFI1 deficiency. Indeed, the inhibition of G1/S progression in Th2 cells by GFI1 deficiency is consistent with previous reports showing the involvement of GFI1 in the progression of the cell cycle in myeloid and lymphoid cells [21]. Other reports also showed that enforced GFI1 expression increases the expression of CDK family genes, such as *Cdk2*, *Cdk4*, *Cdk5*, *Cdk7* and *Cdk8*, which are positive regulators of the cell cycle, especially in relation to G1/S progression, whereas GFI1 suppressed the expression of *Cdkn1b*, *Cdkn1a* and *Cdkn2b*, which are considered to be inhibitors of G1/S progression [30, 31, 35]. Furthermore, we found that IFNγ treatment increased the Cdkn1a expression. We therefore think that the molecular mechanism by which IFNγ mediates inhibition of Th2 cell proliferation is due to a reduced GFI1 expression, which results in the inhibition of G1/S progression by increasing the Cdkn1a expression in Th2 cells.

In conclusion, we demonstrated that the mechanism underlying the IFNγ-mediated inhibition of Th2 cell proliferation involves IFNγ suppressing GFI expression, thereby inhibiting Th2 cell proliferation. This is the one mechanism through which an immune response is amplified toward an appropriate type of response (e.g. Type I) by not only enhancing the specific type of responses (e.g. Type I) but also by inhibiting other types of responses (e.g. Type II).

## Supporting information

**S1 Fig.** (A) The cytokine production of Th2 cells that were *in vitro*-differentiated using polyclonal stimulations, as described in S1B Fig. (B) A schematic illustration of the experimental

protocol using C57BL/6 mice. Naïve CD4 T cells were stimulated with soluble anti-CD3 and anti-CD28 Abs together with T-depleted splenocytes in the presence of IL-4, IL-2 and anti-IFNγ for 3 days, and further cultured with IL-2 for another 2 days to make fully differentiated Th2 cells. Cells were then labelled with CFSE, and further cultured with the indicated cytokines for the indicated number of days. (C) The MFI of CFSE dilution on cells cultured with the indicated concentration of IFNγ treatment for 4 days. (D) A schematic illustration of the experimental protocol using OTII TCR Tg mice. Naïve CD4 T cells were stimulated with OVA peptide (Loh15) together with T-depleted splenocytes in the presence of IL-4, IL-2 and anti-IFNγ for 3 days, and then further cultured with IL-2 for another 2 days to make fully differentiated Th2 cells. Cells were then labelled with CFSE, and further cultured with the indicated cytokines for the indicated number of days. (E) The cytokine production of Th2 cells that were *in vitro*-generated using OVA peptide stimulations as described in S1C Fig (F and G) A histogram (F) and the MFI (G) of CFSE on antigen-specific OTII TCR Tg Th2 cells cultured with IL-2 in the presence or absence of IFNγ for 4 days are shown. (H and I) A cell cycle analysis was performed using antigen-specific OTII TCR Tg Th2 cells after cultivation with IL-2 in the presence or absence of IFNγ for 4 days. (J) The MFI of CD25, CD122 and CD132 on Th2 cells cultured with or without IFNγ for 4 days is shown. Iso, isotype control.
(PDF)

**S2 Fig.** (A) A schematic illustration of the experimental protocol using *Tbx21*−/− mice. Naïve CD4 T cells were isolated from *Tbx21*−/− (CD45.2), *Tbx21*+/+ (CD45.2) and congenic C57BL/6 (CD45.1) mice and cultured under Th2 conditions. *Tbx21*−/− Th2 cells were mixed together with CD45.1 Wt Th2 cells, labelled with CFSE and cultured with IL-2 in the presence of 4-OHT for the indicated number of days. (B) The *Tbx21* mRNA expression relative to *Hprt* on Th2 cells after treatment with or without IFNγ. Error bars represent the mean ± SD. Data are representative of three independent experiments. (C) Intracellular staining of IL-4 and IFNγ production of *Tbx21*+/+ or *Tbx21*−/−CD4 T cells cultured for 5 days under Th1-skewing or Th2-skewing conditions is shown. (D) The MFI of CFSE on Th2 cells after cultivation for the indicated number of days. Data are representative of two independent experiments. (E) A flow cytometry analysis showing the MFI of CFSE dilution in Th1 cells after the cultivation with IFNγ or anti-IFNγ Ab for the indicated days.
(PDF)

**S3 Fig.** (A) *In vitro*-differentiated Th2 cells were further cultured in the presence or absence of IFNγ for 3 days. mRNA was prepared from these cells, and quantitative RT-PCR analysis of the indicated genes was performed. These gene expression changes are shown after being normalized with *Hprt*. The ratio of the gene expressions of IFNγ-treated cells relative to IFNγ-untreated cells) is shown. (B) A comparison of the relative expression ($\log_2$ values) of the selected genes between IFNγ-treated and untreated Th2 cells in (A).
(PDF)

**S4 Fig.** (A) A schematic illustration of the experimental protocol. Naïve CD4 T cells were isolated from *Gfi1*fl/fl CreER[T2] (CD45.2), *Gfi1*+/+ CreER[T2] (CD45.2) and CD45.1 congenic mice and cultured under Th2 conditions. *Gfi1*fl/fl CreER[T2] (KO) or *Gfi1*+/+ CreER[T2] (WT) Th2 cells were mixed together with CD45.1 Wt Th2 cells, labelled with CFSE and further cultured with indicated cytokines (i.e., IL-2 and IFNγ) in the presence of 4-OHT for the indicated number of days. (B) GFI-1 mRNA expression relative to *Hprt* on cells after treatment of 4-OHT for 1 day. (C) The cytokine production of Th2 cells generated from the indicated mice as described in (A). Numbers indicate the percentage in each quadrant. Data are representative of three independent experiments. (D and E) A schematic illustration of the experimental protocol using

retroviral infection. Naïve CD4 T cells were isolated from C57BL/6 mice and cultured under Th2 conditions for 5 days. *In vitro*-differentiated Th2 cells were then labelled with CFSE, infected with the GFI1-Thy1.1-retrovirus (RV) or empty-Thy1.1-RV and further cultured with IL-2 in the presence or absence of IFNγ. The MFI of CFSE was measured on day 3 or 4 by flow cytometry (D). The infection efficiency of Control-Thy1.1 RV and GFI1-Thy1.1 RV (E). (PDF)

**S1 Table. Primer sequence and probes for quantitative RT-PCR.**
(XLS)

## Acknowledgments

We thank K. Sugaya, S. Tetsuyoshi, and T. Nakajima for their excellent technical assistance.

## Author Contributions

**Conceptualization:** Ryoji Yagi, Toshinori Nakayama.

**Data curation:** Murshed H. Sarkar, Ryoji Yagi.

**Funding acquisition:** Ryoji Yagi, Ilkka S. Junttila, Motoko Y. Kimura, Toshinori Nakayama.

**Investigation:** Murshed H. Sarkar, Ryoji Yagi, Yukihiro Endo, Ryo Koyama-Nasu, Yangsong Wang, Ichita Hasegawa, Toshihiro Ito, Ilkka S. Junttila, Jinfang Zhu, Motoko Y. Kimura.

**Project administration:** Ryoji Yagi.

**Supervision:** Ryoji Yagi, Motoko Y. Kimura, Toshinori Nakayama.

**Writing – original draft:** Ryoji Yagi, Motoko Y. Kimura.

**Writing – review & editing:** Motoko Y. Kimura.

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
