## [Decision Letter · Decision Letter 0]

26 Feb 2021

PONE-D-20-39447

IFNγ suppresses the expression of GFI1 and thereby inhibits Th2 cell proliferation

PLOS ONE

Dear Dr. Kimura,

Thank you for submitting your manuscript to PLOS ONE. After careful consideration, we feel that it has merit but does not fully meet PLOS ONE’s publication criteria as it currently stands. Therefore, we invite you to submit a revised version of the manuscript that addresses the points raised during the review process.

We look forward to receiving your revised manuscript.

Kind regards,

Charalampos G Spilianakis, Ph.D

Academic Editor

PLOS ONE

Reviewers' comments:

Reviewer's Responses to Questions

**Comments to the Author**

1. Is the manuscript technically sound, and do the data support the conclusions?

Reviewer #1: Partly

Reviewer #2: Yes

Reviewer #3: Partly

Reviewer #4: Yes

2. Has the statistical analysis been performed appropriately and rigorously? 

Reviewer #1: Yes

Reviewer #2: Yes

Reviewer #3: Yes

Reviewer #4: Yes

3. Have the authors made all data underlying the findings in their manuscript fully available?

Reviewer #1: Yes

Reviewer #2: Yes

Reviewer #3: Yes

Reviewer #4: Yes

4. Is the manuscript presented in an intelligible fashion and written in standard English?

Reviewer #1: Yes

Reviewer #2: Yes

Reviewer #3: Yes

Reviewer #4: Yes

5. Review Comments to the Author

Reviewer #1: Comments on manuscript PONE 20_39447

The manuscript entitled “IFNγ suppresses the expression of GFI1 and thereby inhibits Th2 cell proliferation” submitted by Sarkar and colleagues aimed to investigate the molecular mechanism by which IFN-gamma inhibits Th2 cell proliferation. The experiments in the study included murine CD4 T cells culture isolated from lymph nodes or spleen. Th2-polarizing conditions were obtained in the presence of IL-4, IL-2 and IFN-gamma. The authors demonstrate that IFN-gamma treatment inhibits Th2 proliferation as measured by CFSE labeling, and further confirm by a coupled immunofluorescence assay with Brdu ad 7-AAD. To identify the molecules that are responsible for the IFN-gamma-mediated inhibition of Th2 cell proliferation, the authors ruled out the participation of the transcription factor T-bet by showing data that proliferation of Th2 cells differentiated from T-bet KO mice was comparable with the wild-type. Then, the authors reasoned GFI1 as a candidate molecule because of its abundant expression in Th2 cells, besides playing a role in Th2 cell proliferation. Further experimentation was performed with differentiated Th2 cells derived from conditional Gfi1-KO mice, and also by retroviral complementation.

- Overall, the manuscript is well written. The work is methodologically sound and the authors performed proper experimentation, although I feel that some important experimental controls and evidence are lacking in some experiments (please, see my comments below).

- Authors should provide more detailed text description about Gfi1 (gene features, protein size, tissue expression, expression regulation, biological function, mechanism of action, animal models available and phenotypes, etc.) in the introduction section. The authors should also discuss the molecular mechanisms by which GFI1 mediates the IFN-gamma-mediated inhibition of Th2 cell proliferation.

- Fig. 2A: authors should extend these analyses by performing RT-qPCR experiments to include expression data of other related (and non-related) genes, especially GFI1 target genes and make correlations with the main idea of the manuscript.

- p. 9: The authors mention that “IFNgamma-mediated downregulation of GFI1 might result in reduced Th2 cell proliferation”. What is the basis for this idea?

- It is important that the authors provide data confirming the null expression of T-bet in Tbx21-/- mice.

- The authors mentioned that “The efficiency of retrovirus infection with the GFI1-retrovirus and control Thy1.1-retrovirus was almost identical”. However, the authors fail to provide direct evidence of GFI1 ectopic expression on infected cells. Authors should provide data confirming the levels of ectopic GFI1 by performing western-blot or IF experiment and comparing with the empty vector to ascertain that GFI1 was efficiently expressed.

- Discussion section in its current format is poor, and it should be definitely improved. Material and methods section could also be improved by providing more detailed description of the procedures and tools generated.

Reviewer #2: This is a straightforward and convincing paper demonstrating that IFNg inhibits TH-2 T cell proliferation by suppressing GFI1 expression. The data is well presented as they use GFI1 KO mice followed by overexpression and they demonstrate the effects of IFNg are consistent with the loss of GFI1 expression. There are only a few points that need clarification

1. Can it be assumed that IFNg was used at 10 ng/ml throughout the study? This was not clearly stated. If so, it is a minor point but was inhibition also seen at 1 ng/ml?

2. It would strengthen the paper to know if inhibition was observed at the RNA level and if so, is the a STAT1 or STAT3 site in the GFI1 gene?

Reviewer #3: The manuscript entitled “IFNγ suppresses the expression of GFI1 and thereby inhibits Th2 cell proliferation” by Sarkar et al. demonstrated that both Th2 cell proliferation and Gfi1 expression were suppressed by IFNγ treatment. The enforced expression of Gfi1 in Th2 cells by retrovirus vector antagonized IFNγ-mediated suppression of Th2 cell proliferation. Thus, they concluded that GFI1 plays a key role in the IFNγ-mediated suppression of Th2 cell proliferation.

The experiments were conducted well and results were potentially important to understand the mechanism IFNγ-mediated suppression of type 2 immune response. However, a series of additional experiments would be required to drawn conclusion. First of all, the molecular mechanism by which IFNγ inhibits Gfi1 expression is not elucidated. Secondly, in vitro-differentiated Th2 cells have functional heterogeneity (heterogeneity in Th2 cytokine production) as shown in Supplementary Figure 1D and 3B. It is possible that the effect of IFNγ is different among Th2 cell subpopulations defined by the ability for Th2 cytokine production. Therefore, the heterogeneity of Th2 cells makes it difficult to draw a conclusion. In addition, the role of Gfi1 on IL-2-dependent proliferation of activated CD4 T cells should be assessed. Thirdly, the impact of Gfi1 deficiency on Th2 cell survival was no determined.

Specific points;

1, The authors also showed that IFNγ suppresses IL-2-dependent proliferation of Th2 cells via inhibition of Gfi1 expression. Although the authors indicated that T-bet was not required for IFNγ-mediated suppression of Th2 cell proliferation (Supplementary Figure 2), the molecular mechanism by which IFNγ inhibits Gfi1 expression remains to be elucidated. It is well known that IFNγ activates Stat1/Stat2 and Stat3 in T cells. Is the activation of these Stats by IFNγ involved in the inhibition of Gfi1 expression? In addition, the impact of IFNγ treatment on the epigenetic status of Gfi1 locus in Th2 cells should be addressed.

2, The in vitro-differentiated Th2 cells contains included both IL-4-producing and-non-producing cells. In addition, some in vitro-differentiated Th2 cells produce IL-13, and other cells produce IL-5 and IL-13. These data suggesting the heterogeneity of in vitro-differentiated Th2 cells. The authors should determine the difference in the expression of Gfi1 among these Th2 cell subpopulations (e.g. IL-4-producing cells, IL-4-non-producing cells, IL-13-producing cells, IL-5/IL-13-double producing cells) and examine the anti-proliferative effect of IFNγ on subpopulations independently.

3, It is important to assess the effect of IFNγ on IL-2 receptor expression in Th2 cells.

4, The activated CD4 T cells expresses Gfi1 and proliferates IL-2-dependent manner. Doses IFNγ has inhibitory effect on IL-2-dependent proliferation of activated CD4 T cells?

5, The effect of IFNγ on the cell death of Th2 cell should be determined.

Reviewer #4: In this study, the author found that IFN� inhibited TH2 proliferation by decreasing Gfi1 expression. This interesting study needs some concerns to be addressed before publication.

- The data on figure 1 indicate a decrease in the frequency of cells in the S phase and the authors conclude this is due to a blockade of G1/S progression. To confirm these data the authors have to analyze cell cycle by the incorporation of propidium iodide. In addition, a more detailed discussion of the increase in G2-M observed have to be provided.

- The authors have to test cell proliferation by additional methods such as cell counting.

- GFI1 expression have to be determined also by western blotting.

- The loss of GFI1 after treatment with 4-OHT have to be showed. In the same way, the increased in GFI1 after infection with retrovirus have to be also showed.

6. PLOS authors have the option to publish the peer review history of their article (what does this mean?). If published, this will include your full peer review and any attached files.

Reviewer #1: No

Reviewer #2: **Yes: **Howard Young

Reviewer #3: No

Reviewer #4: No

---

## [Author Response · Author response to Decision Letter 0]

28 Sep 2021

Point-by-point responses (Manuscript PONE 20_39447)

We very much appreciate the efforts of all reviewers and appreciate their valuable comments and suggestions for improving our manuscript. 

Reviewer #1: 

The manuscript entitled “IFNγ suppresses the expression of GFI1 and thereby inhibits Th2 cell proliferation” submitted by Sarkar and colleagues aimed to investigate the molecular mechanism by which IFN-gamma inhibits Th2 cell proliferation. The experiments in the study included murine CD4 T cells culture isolated from lymph nodes or spleen. Th2-polarizing conditions were obtained in the presence of IL-4, IL-2 and IFN-gamma. The authors demonstrate that IFN-gamma treatment inhibits Th2 proliferation as measured by CFSE labeling, and further confirm by a coupled immunofluorescence assay with Brdu ad 7-AAD. To identify the molecules that are responsible for the IFN-gamma-mediated inhibition of Th2 cell proliferation, the authors ruled out the participation of the transcription factor T-bet by showing data that proliferation of Th2 cells differentiated from T-bet KO mice was comparable with the wild-type. Then, the authors reasoned GFI1 as a candidate molecule because of its abundant expression in Th2 cells, besides playing a role in Th2 cell proliferation. Further experimentation was performed with differentiated Th2 cells derived from conditional Gfi1-KO mice, and also by retroviral complementation.

- Overall, the manuscript is well written. The work is methodologically sound and the authors performed proper experimentation, although I feel that some important experimental controls and evidence are lacking in some experiments (please, see my comments below).

Our response:

Thank you very much for your careful reading our manuscript. 

- Authors should provide more detailed text description about Gfi1 (gene features, protein size, tissue expression, expression regulation, biological function, mechanism of action, animal models available and phenotypes, etc.) in the introduction section. The authors should also discuss the molecular mechanisms by which GFI1 mediates the IFN-gamma-mediated inhibition of Th2 cell proliferation.

Our response: 

As suggested, we added more detailed information concerning Gfi1 to the introduction section (highlighted in red characters) in p4 line 8 ~ p4 line 16. We also added more discussion regarding the possible molecular mechanisms by which GFI1 regulates the IFNγ-mediated inhibition of Th2 cell proliferation in the discussion (highlighted in red characters) in p16 line 10 ~ p16 line 16. 

- Fig. 2A: authors should extend these analyses by performing RT-qPCR experiments to include expression data of other related (and non-related) genes, especially GFI1 target genes and make correlations with the main idea of the manuscript.

Our response: 

To identify the molecules responsible for the IFNγ-mediated inhibition of Th2 cell proliferation, we performed a RT-PCR analysis of 37 transcription factors reported to be involved in the cell growth, development, effector function, apoptosis or survival of helper T cells. We have now added these data to new Supplementary Figures 3A and 3B and a description to the text (highlighted in red characters) in p11 line 9 ~ p11 line 20. In addition, we performed RT-PCR to examine the expression of Cdkn1a, a cell cycle inhibitor known to be negatively regulated by the GFI1 target gene. We found that IFNγ treatment increased the mRNA expression of Cdkn1. We have now added these data to the new Figure 2B and a description to the text (highlighted in red characters) in p11 line 21 ~ p11 line 23. 

- p. 9: The authors mention that “IFNgamma-mediated downregulation of GFI1 might result in reduced Th2 cell proliferation”. What is the basis for this idea?

Our response:

Thank you very much for your question. Since our data showed that IFN� suppressed both the GFI1 expression and Th2 cell proliferation and that GFI1 inhibits Cdkn1 (a cell cycle inhibitor) expression (new Figure 2B), we raised a possibility that the reduction in Th2 cells proliferation might have ben dependent on the decreased expression of GFI1. This is a hypothesis based on our experimental data. 

- It is important that the authors provide data confirming the null expression of T-bet in Tbx21-/- mice.

Our response: 

The Tbx21-/- mouse that we used in this study is well established (Szabo et al. Science 295 338-342, 2002, DOI: 10.1126/science.1065543). In fact, we confirmed that the Th1 cell differentiation using CD4T cells from Tbx21-/- mice was significantly diminished, indicating that the CD4 T cells lack T-bet expression. We have newly added supplementary Figure 2C and included a description in the text (highlighted in red characters) in p10 line 23 ~ p11 line 1.

- The authors mentioned that “The efficiency of retrovirus infection with the GFI1-retrovirus and control Thy1.1-retrovirus was almost identical”. However, the authors fail to provide direct evidence of GFI1 ectopic expression on infected cells. Authors should provide data confirming the levels of ectopic GFI1 by performing western-blot or IF experiment and comparing with the empty vector to ascertain that GFI1 was efficiently expressed.

Our response: 

We attempted intracellular staining and Western blotting to detect GFI1 protein; however, we were unsuccessful, probably due to the nature of the commercially available Ab used. We used three kinds of Abs from SANTA CRUIZ BIOTECHNOLOGY, INC (sc-101053, sc-373960, and sc-376949), but the Abs unfortunately it did not work with the primary cultured cells, at least in our hands. We confirmed that our GFI1-retrovirus was functional by quantitative RT-PCR. We detected GFI-1 mRNA in GFI1 retrovirally infected GFI1-deficient CD4T cells from GFI1fl/flxCD4CreTg mice but did not detect it in control-infected GFI1-deficient CD4T cells (Please see Figure for reviewers). We hope that this reviewer agrees that the mRNA expression and the experimental data shown in the new Figure 3E is evident to show that overexpression went well. 

- Discussion section in its current format is poor, and it should be definitely improved. Material and methods section could also be improved by providing more detailed description of the procedures and tools generated.

Our response: 

As requested, we revised the Discussion section (highlighted in red characters) as well as the Materials and methods section, providing more detailed information (highlighted in red characters) We hope that these changes now satisfy this reviewer.

Reviewer #2: 

This is a straightforward and convincing paper demonstrating that IFNg inhibits TH-2 T cell proliferation by suppressing GFI1 expression. The data is well presented as they use GFI1 KO mice followed by overexpression and they demonstrate the effects of IFNg are consistent with the loss of GFI1 expression. There are only a few points that need clarification

1. Can it be assumed that IFNg was used at 10 ng/ml throughout the study? This was not clearly stated. If so, it is a minor point but was inhibition also seen at 1 ng/ml?

Our response: 

We used 10 ng/ml of IFNγ throughout the study. We have now mentioned this in the Materials and Methods section (highlighted in red characters) in p6 line 6 ~ line 8). In addition, as requested, we performed a new experiment using different concentrations of IFNγ (0.3, 1, 3, 10 and 30 ng/ml). We found that the suppressive effect of IFNγ on Th2 cell proliferation was observed even at 1 ng/ml, although the impact was slightly weaker than when using 10 ng/ml. We have now added these data to the new Supplementary Figure 1C and the revised text (highlighted in red characters) in p9 line 11 ~ p9 line 15) 

2. It would strengthen the paper to know if inhibition was observed at the RNA level and if so, is the a STAT1 or STAT3 site in the GFI1 gene?

Our response

As shown in Figure 2A, the inhibitory effect by IFNγ on GFI1 occurs at the RNA level. As requested, we investigated the binding of STATs to the GFI1 gene and found that there were two STAT binding sites within intron 1. In addition, a previous report showed that STAT3-deficiency (by siRNA) results in an increased GFI1 mRNA expression, indicating that STAT3 activation inhibits GFI1 expression (Tripathi et al., Cell Reports, 2017). 

Reviewer #3: 

The manuscript entitled “IFNγ suppresses the expression of GFI1 and thereby inhibits Th2 cell proliferation” by Sarkar et al. demonstrated that both Th2 cell proliferation and Gfi1 expression were suppressed by IFNγ treatment. The enforced expression of Gfi1 in Th2 cells by retrovirus vector antagonized IFNγ-mediated suppression of Th2 cell proliferation. Thus, they concluded that GFI1 plays a key role in the IFNγ-mediated suppression of Th2 cell proliferation.

Our response:

Thank you very much for your careful reading our manuscript. 

The experiments were conducted well and results were potentially important to understand the mechanism IFNγ-mediated suppression of type 2 immune response. However, a series of additional experiments would be required to drawn conclusion. First of all, the molecular mechanism by which IFNγ inhibits Gfi1 expression is not elucidated. Secondly, in vitro-differentiated Th2 cells have functional heterogeneity (heterogeneity in Th2 cytokine production) as shown in Supplementary Figure 1D and 3B. It is possible that the effect of IFNγ is different among Th2 cell subpopulations defined by the ability for Th2 cytokine production. Therefore, the heterogeneity of Th2 cells makes it difficult to draw a conclusion. In addition, the role of Gfi1 on IL-2-dependent proliferation of activated CD4 T cells should be assessed. Thirdly, the impact of Gfi1 deficiency on Th2 cell survival was no determined.

Our response:

Thank you very much for your careful reading and thoughtful, constructive critique. We have now performed additional experiments, added several new Figures and added description to the text, as described below. 

Specific points;

1, The authors also showed that IFNγ suppresses IL-2-dependent proliferation of Th2 cells via inhibition of Gfi1 expression. Although the authors indicated that T-bet was not required for IFNγ-mediated suppression of Th2 cell proliferation (Supplementary Figure 2), the molecular mechanism by which IFNγ inhibits Gfi1 expression remains to be elucidated. It is well known that IFNγ activates Stat1/Stat2 and Stat3 in T cells. Is the activation of these Stats by IFNγ involved in the inhibition of Gfi1 expression? In addition, the impact of IFNγ treatment on the epigenetic status of Gfi1 locus in Th2 cells should be addressed.

Our response: 

Regarding the molecular mechanism by which IFNγ inhibits Gfi1 expression, we performed experiments using Stat1 inhibitor (Fludarabine) to determine whether STAT1 inhibition rescued the IFNγ-mediated inhibition of Th2 cell proliferation. Interestingly, while we did not observe the restoration of proliferation, as the MFI of CFSE on IFNγ-treated Th2 cells did not change between the presence or absence of Fludarabine (new Fig. 3D left), we did detect some restoration of the number of IFNγ-treated cells in the presence of Fludarabine (new Fig. 3D right). These data suggest that the IFNγ-mediated inhibition of Th2 cell proliferation was not regulated through STAT1 activation. We have now newly added Figure 3D and described these findings in the text (highlighted in red characters) in p13 line16 ~ p13 line23. 

Regarding the impact of IFNγ treatment on the epigenetic status of Gfi1 locus in Th2 cells, this is beyond the scope of our study, as we did not analyze the epigenetic status of GFI1 in Th2 cells. Such an experiment will be performed in a future study. 

2. The in vitro-differentiated Th2 cells contains included both IL-4-producing and-non-producing cells. In addition, some in vitro-differentiated Th2 cells produce IL-13, and other cells produce IL-5 and IL-13. These data suggesting the heterogeneity of in vitro-differentiated Th2 cells. The authors should determine the difference in the expression of Gfi1 among these Th2 cell subpopulations (e.g. IL-4-producing cells, IL-4-non-producing cells, IL-13-producing cells, IL-5/IL-13-double producing cells) and examine the anti-proliferative effect of IFNγ on subpopulations independently.

Our response: 

It has been reported that in vitro-differentiated IL-4-producing (IL-4+) and IL-4-non-producing (IL-4–) Th2 cells express similar levels of GATA3 mRNA, a transcription factor critical for Th2 cells. Furthermore, IL-4+ and IL-4– Th2 cells have a similar potential for IL-4 production upon subsequent stimulation (Hu-Li et al., Immunity 2001). Accordingly, we do not feel that the experiment suggested here is worth performing. 

Nevertheless, we tried to perform GFI1 staining via intracellular staining, since the experiment requires a single cell analysis. However, our efforts were not successful, probably due to the nature of the commercially available Ab used.

3, It is important to assess the effect of IFNγ on IL-2 receptor expression in Th2 cells.

Our response: 

As requested, we examined the expression levels of CD25 (IL-2R�), CD122 (IL-2R�) and CD132 (�C). The expression level of CD132 was comparable between IFN�-treated and untreated cells, whereas the CD25 and CD122 expressions was rather increased by IFN� treatment. These data indicate that IFNγ treatment did not dampen the IL-2 receptor expression in Th2 cells, and the low proliferation is not due to a low expression of IL-2 receptor complex. We have now newly added supplementary Figure 1J and described our findings in the text (highlighted in red characters) in p10 line 7 ~ line 12.

4, The activated CD4 T cells express Gfi1 and proliferates IL-2-dependent manner. Dose IFNγ has inhibitory effect on IL-2-dependent proliferation of activated CD4 T cells?

Our response: 

As requested, we performed the experiment using Th1 cells, which are known to be dependent on IL-2 for proliferation. We found that Th1 cells did not show any proliferative defects following IFNγ treatment, indicating that IFNγ did not influence IL-2-dependent Th1 cell proliferation. We have newly added supplementary Figure 2E and described our findings in the text (highlighted in red characters) in p11 line 3 ~ line 4. 

5, The effect of IFNγ on the cell death of Th2 cell should be determined.

Our response: 

As requested, we performed experiments to examine the effect of IFNγ on the death of Th2 cells. We have newly added Figure 1F and described our findings in the text (highlighted in red characters) in p10 line 1 ~ p10 line 5.

Reviewer #4: 

In this study, the author found that IFN� inhibited TH2 proliferation by decreasing Gfi1 expression. This interesting study needs some concerns to be addressed before publication.

Our response:

Thank you very much for your careful reading our manuscript. 

- The data on figure 1 indicate a decrease in the frequency of cells in the S phase and the authors conclude this is due to a blockade of G1/S progression. To confirm these data the authors have to analyze cell cycle by the incorporation of propidium iodide. In addition, a more detailed discussion of the increase in G2-M observed have to be provided.

Our response: more discussion required

We used 7AAD instead of PI because 7AAD is known to have less spectral overlap than PI in flow cytometry, which prevented any possible errors due to multiple staining. Furthermore, 7AAD is more stable than PI and does not leach from cells (Ref. Falzone et al, Andrologia. 2010, doi: 10.1111/j.1439-0272.2009.00949.x.). We hope that the 7AAD data we provided satisfy this reviewer. 

Regarding the increase in G2-M observed, we added more detailed discussion to the text (highlighted in red characters) in p15 line 24 ~ p16 line 5). 

- The authors have to test cell proliferation by additional methods such as cell counting.

Our response: 

We have now added new data showing the actual cell number after in vitro culture with IFNγ. We newly added Figure 1C and described our findings in the text (highlighted in red characters) in p9 line 8 ~ line 9. 

- GFI1 expression have to be determined also by western blotting.

Our response: 

As requested, we tried the Western blotting to detect GFI1 protein; however, we were unsuccessful, probably due to the nature of the commercially available Ab used. We used three kinds of Abs from Santa Cruz Biotechnology, Inc. (sc-101053, sc-373960, and sc-376949), but the Abs unfortunately did not work with the primary cultured cells, at least in our experience. We hope that this reviewer feels that the mRNA expression shown in Figure 2A is sufficient to prove that IFNγ treatment decreases the GFI1 expression. 

- The loss of GFI1 after treatment with 4-OHT have to be showed. In the same way, the increased in GFI1 after infection with retrovirus have to be also showed.

Our response:

As requested, we tried the Western blotting to detect GFI1 protein; however, we were unsuccessful, probably due to the nature of the commercially available Ab used. We used three kinds of Abs from Santa Cruz Biotechnology, Inc. (sc-101053, sc-373960, and sc-376949), but the Abs unfortunately did not work with the primary cultured cells, at least in our experience. Instead, we performed RT-PCR to detect the GFI1 mRNA expression of cells after treatment with 4-OHT (new Supplemental figure 4B), and described in the text (highlighted in red characters) in p12 line 4 ~ line6). The GFI1 mRNA level was significantly reduced in cells with 4-OHT treatment. We also confirmed that our GFI1-retrovirus was functional by quantitative RT-PCR. We detected GFI-1 mRNA in GFI1 retrovirally infected GFI1-deficient CD4T cells from GFI1fl/flxCD4CreTg mice but did not detect it in control-GFI1-deficient CD4T cells (Please see Figure for reviewers). We hope that this reviewer agrees that the mRNA expression and experimental data shown in Figure 2 and 3 is evident to show that the deletion and overexpression went well.

---

## [Decision Letter · Decision Letter 1]

5 Nov 2021

IFNγ suppresses the expression of GFI1 and thereby inhibits Th2 cell proliferation

PONE-D-20-39447R1

Dear Dr. Kimura,

We’re pleased to inform you that your manuscript has been judged scientifically suitable for publication and will be formally accepted for publication once it meets all outstanding technical requirements.

Kind regards,

Charalampos G Spilianakis, Ph.D

Academic Editor

PLOS ONE

Additional Editor Comments (optional):

Reviewers' comments:

Reviewer's Responses to Questions

**Comments to the Author**

1. If the authors have adequately addressed your comments raised in a previous round of review and you feel that this manuscript is now acceptable for publication, you may indicate that here to bypass the “Comments to the Author” section, enter your conflict of interest statement in the “Confidential to Editor” section, and submit your "Accept" recommendation.

Reviewer #1: All comments have been addressed

Reviewer #2: All comments have been addressed

2. Is the manuscript technically sound, and do the data support the conclusions?

Reviewer #1: Yes

Reviewer #2: Yes

3. Has the statistical analysis been performed appropriately and rigorously? 

Reviewer #1: Yes

Reviewer #2: Yes

4. Have the authors made all data underlying the findings in their manuscript fully available?

Reviewer #1: Yes

Reviewer #2: Yes

5. Is the manuscript presented in an intelligible fashion and written in standard English?

Reviewer #1: Yes

Reviewer #2: Yes

6. Review Comments to the Author

Reviewer #1: (No Response)

Reviewer #2: My previous concerns have been fully addressed. I have no further issues that need to be addressed.

7. PLOS authors have the option to publish the peer review history of their article (what does this mean?). If published, this will include your full peer review and any attached files.

Reviewer #1: No

Reviewer #2: **Yes: **Howard A Young

---

## [Editor Report · Acceptance letter]

12 Nov 2021

PONE-D-20-39447R1 

IFNγ suppresses the expression of GFI1 and thereby inhibits Th2 cell proliferation 

Dear Dr. Kimura:

I'm pleased to inform you that your manuscript has been deemed suitable for publication in PLOS ONE. Congratulations! Your manuscript is now with our production department. 

Kind regards, 

on behalf of

Dr. Charalampos G Spilianakis 

Academic Editor

PLOS ONE